# Comprehensive Attention Self-Distillation for Weakly-Supervised Object Detection

**Zeyi Huang**[*], **Yang Zou**,[*] **Vijayakumar Bhagavatula, and Dong Huang**
Carnegie Mellon University, Pittsburgh, PA, 15213, USA
{zeyih@andrew, yzou2@andrew, kumar@ece, donghuang@}.cmu.edu

## Abstract

Weakly Supervised Object Detection (WSOD) has emerged as an effective tool to train object detectors using only the image-level category labels. However, without object-level labels, WSOD detectors are prone to detect bounding boxes on salient objects, clustered objects and discriminative object parts. Moreover, the image-level category labels do not enforce consistent object detection across different transformations of the same images. To address the above issues, we propose a Comprehensive Attention Self-Distillation (CASD) training approach[2] for WSOD. To balance feature learning among all object instances, CASD computes the comprehensive attention aggregated from multiple transformations and feature layers of the same images. To enforce consistent spatial supervision on objects, CASD conducts self-distillation on the WSOD networks, such that the comprehensive attention is approximated simultaneously by multiple transformations and feature layers of the same images. CASD produces new state-of-the-art WSOD results on standard benchmarks such as PASCAL VOC 2007/2012 and MS-COCO.

## 1  Introduction

Visual object detection has achieved remarkable progress in the last decade thanks to the advances of Convolutional Neural Networks (CNNs) [1, 2]. An integral part of the achievement is the availability of large-scale training data with precise bounding-box annotations (PASCAL VOC [3], MS-COCO [4], etc). However, obtaining such fine-grained annotations at a large scale is labor-intensive and time-consuming, which drove many researchers to explore the weakly-supervised setting. Weakly-Supervised Object Detection (WSOD) [5] aims to learn object detectors with only the image-level category labels indicating whether an image contains an object or not.

Most previous methods for WSOD are based on the Multiple Instance Learning (MIL) [6]. These methods regard images as bags and object proposals as instances. A positive bag contains at least one positive instance while all instances being negative in a negative bag. WSOD instance classifiers (object detectors) are trained over these bags. Recently, leveraging the powerful representation learning capacity of CNNs, several researchers proposed end-to-end MIL networks (OICR [7], PCL [7], MIST [8], [9, 10]) with promising WSOD performances. These CNN methods regard the instance classification (object detection) problem as a latent model learning within a bag classification (image classification) problem, where the final image scores are the aggregation of the instance scores. However, due to the under-determined and ill-posed nature of WSOD, there is still a large performance gap between the weakly-supervised detectors and fully-supervised detectors.

The existing methods have two main sets of issues as demonstrated in Fig. 1. First, in the "Biased WSOD" column of Fig. 1 (a) , there are three typical problems. *Missing instance*: Salient objects

---

[*]The authors contributed equally.

[2]Code are avaliable at https://github.com/DeLightCMU/CASD

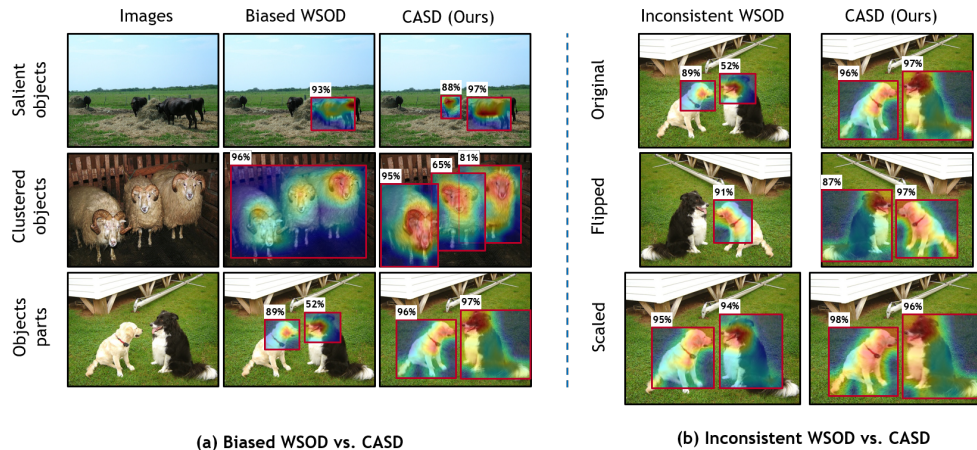

Figure 1: Typical WSOD issues and our CASD solution. **Only the attention maps of high confidence proposals by WSOD detection are overlaid on input images.** (a) Biased WSOD detects salient objects, clustered instances and object parts. (b) Inconsistent WSOD detects different objects on different transformations of the same image.

are easily detected while inconspicuous instances tend to be ignored. *Clustered instances*: multiple adjacent instances of the same category may be detected in a single bounding box. *Part domination:* The bounding boxes are prone to focus on the most discriminative object parts instead of the entire objects. Second, in the "Inconsistent WSOD" column of Fig. 1 (b), the same image and its different image transformations, i.e., "Original Image", "Flipped Image" and "Scaled Image", do not produce the same object bounding boxes.

WSOD conducts classification on object proposals (e.g., bounding boxes generated by selective search [11]) with image-level class labels. The object proposals receive high classification scores are considered as objects detected by WSOD. As we dive deep into the above issues from a feature learning perspective, we overlay the attention maps of object proposals that get high confidences in WSOD (Fig. 1). High intensity in attention maps corresponds to highly discrimiative and biased features learned by the WSOD networks. We observe the drawbacks of WSOD detection are closely associated with the issues in feature learning. For "Biased WSOD", it is clear that salient objects, clustered objects, and certain object parts contain spatial features that dominate the WSOD classification. From a statistical machine learning point-of-view, feature domination is typically established by the biased feature distribution in training data. For "Inconsistent WSOD", the different transformations of the same image are typically generated by data augmentation and are used to train the WSOD networks in different training iterations. The same class image-level labels of transformed images do not enforce spatially consistent feature learning and may lead to part domination and missing instances. Note that the inconsistency on feature localization was not an issue for full-supervised setting where augmented training data with precise bounding box labels can naturally encourage consistency.

The above observations inspire us to address WSOD issues using an attention-based feature learning method. We propose a Comprehensive Attention Self-Distillation (CASD) approach for WSOD training. To balance feature learning among objects, CASD computes the comprehensive attention aggregated from multiple transformations and feature layers of the same images. The "CASD (ours)" column of Fig. 1 (a) demonstrates that CASD generates balanced attention on less salient objects, individual objects, and entire objects, which enables WSOD detection on these objects. To enforce consistent spatial supervision on objects, CASD conducts self-distillation on the WSOD network itself, such that the comprehensive attention is approximated simultaneously by multiple transformations and layers of the same images. The "CASD (ours)" column of Fig. 1 (b) demonstrates that CASD generates consistent attention on different transformed variants of the same image, leading to consistent WSOD detection in different transformations.

By computing the comprehensive attention maps, CASD aggregates "free" resources of spatial supervision for WSOD, including image transformations and low-to-high feature layers. By conducting self-distillation on the WSOD network with the comprehensive attention maps, CASD enforces

instance-balanced and spatially-consistent supervision, therefore robust bounding box localization for WSOD. CASD achieves the state-of-the-art on several standard benchmarks, e.g. PASCAL VOC 2007/2012 and MS-COCO, outperforming other methods by clear margins. Systematic ablation studies are also conducted on the effects of transformations and feature layers on CASD.

## 2 Related Work

### 2.1 Weakly Supervised Object Detection

Recent WSOD performance are significantly boosted by incorporating Multiple Instance Learning (MIL) in Convolutional Neural Networks (CNN). [5] introduces the first end-to-end Weakly Supervised Deep Detection Network (WSDDN) with MIL, which inspired many following works. [12] combines WSDDN and multi-stage instance classifiers into an Online Instance Classifier Refinement (OICR) framework. [7] further improves OICR with a robust proposal generation module based on proposal clustering, namely Proposal Cluster Learning (PCL). [13] introduces Continuation Multiple Instance Learning (C-MIL) by relaxing the original MIL loss function with a set of smoothed loss functions preventing detectors to be part dominating. [8] proposes a multiple instance self-training framework with an online regression branch. [14] and [15] leverage segmentation maps to generate instance proposals with rich contextual information. [16] and [17] introduce detection-segmentation cyclic collaborative frameworks. Different from the above methods that regularize the WSOD outputs, CASD directly enforces comprehensive and consistent WSOD feature learning.

### 2.2 Attention Mechanism in Computer Vision DNNs

The attention mechanism provides a fine-grained view of the features learned in CNNs. [18, 19] introduce the connection between class-wise attention maps and image-level class labels. Due to its explicit spatial clues, attention maps have been used to improve computer vision tasks in two ways: (1) **Re-weighting features.** [20, 21, 22, 23, 24, 25] re-weight features with spatial-wise attention maps. [26, 27, 28] improve supervised tasks by inverting gradient-based spatial-wise and channel-wise attention. (2) **Loss regularization.** [29] introduces a consistency loss between attention maps under different input transformations for image classification. [30, 31] propose cross-layer consistency losses over attention maps for image classification and lane detection. To the best of our knowledge, CASD is the first attempt to explore attention regularization for WSOD. Moreover, existing methods for other tasks only encourage consistency of features, rather than the completeness of features. Specifically, the features tend to focus on object parts but fail to localize less salient objects in WSOD. CASD encourages both consistent and spatially complete feature learning guided by the comprehensive attention maps, which explicitly addresses WSOD issues above.

### 2.3 Knowledge Distillation

Knowledge distillation [32, 33] is a CNN training process to transfer knowledge from teacher networks to student networks. It has found wide applications in model compression, incremental learning, and continual learning. The student networks mimic the the teacher networks on predictive probabilities [32, 34], intermediate features [35], or attention maps of intermediate neural activations [36, 37, 38, 39]. In contrast, CASD is a knowledge *self-distillation* process that transfers the comprehensive attention knowledge within the WSOD model itself, rather than another teacher model, across multiple views of the same data.

## 3 Method

### 3.1 Background

We first review a basic WSOD framework called Online Instance Classifier Refinement (OICR) [12], then introduce CASD as a general training module on top of OICR. Formally, we denote $\mathbf{I} \in \mathbb{R}^{h \times w \times 3}$ as an RGB image and $\mathbf{y} = [y_1, ..., y_C] \in [0, 1]^C$ as the labels associated with $\mathbf{I}$. $C$ is the total number of object categories. $\mathbf{y}$ is a binary vector where $y_c = 1$ indicates the presence of at least one object of the $c_{th}$ category and $y_c = 0$ indicates absence. The OICR framework consists of two main components (See Fig. 2):

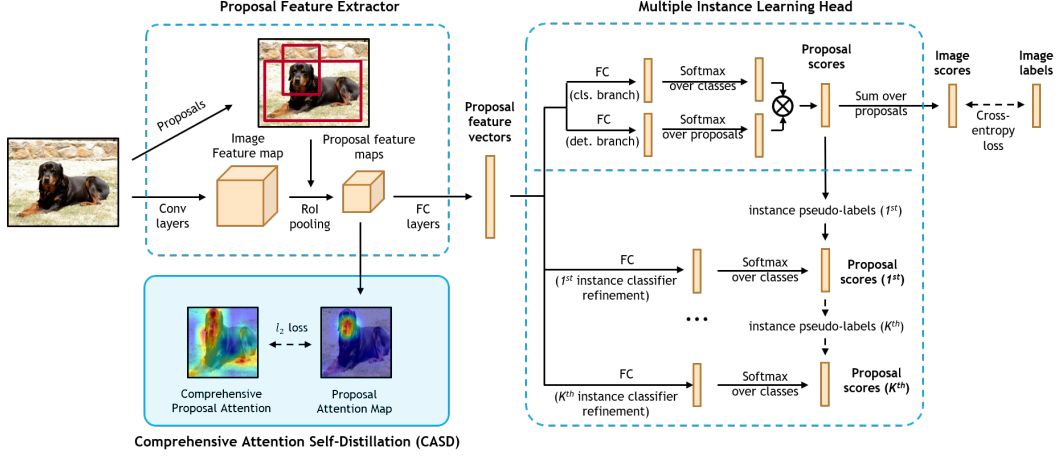

Figure 2: The OICR WSOD network [12] with our Comprehensive Attention Self-Distillation (CASD) training module. OICR includes two parts: **Proposal Feature Extractor** extracts feature vectors for proposals. **Multiple Instance Learning Head** aggregates the proposal scores for image-level classification. $K$ branches of instance classifiers are sequentially refined for better localization. CASD computes proposal attention maps from each proposal feature maps, aggregates attentions into comprehensive attention maps, and self-distills the comprehensive attention maps to improve WSOD (described in Sec. 3.2).

**Proposal Feature Extractor**. From an input image $\mathbf{I}$, the object proposals $\mathbf{R} = \{R_1, R_2, ..., R_N\}$ are generated by Selective Search [11] where $N$ is total number of proposals (bounding boxes). Then, a CNN backbone is used to extract the image feature maps for $\mathbf{I}$, from which the proposal feature maps are extracted respectively. Lastly, these proposal feature maps are fed to a Region-of-Interest (RoI) Pooling layer [1] and two Fully-Connected (FC) layers to obtain proposal feature vectors.

**Multiple Instance Learning (MIL) Head.** The MIL head learns the latent instance classifiers (detectors). This module takes the above-obtained proposal feature vectors as input and conducts the image-level MIL classification, regarding detection as a latent model learning problem.

Following WSDDN [5], the proposal feature vectors are forked into two parallel classification and detection branches to generate two matrices $\mathbf{x}^{cls}, \mathbf{x}^{det} \in \mathbb{R}^{C \times N}$. Each column of $\mathbf{x}^{cls}$ and $\mathbf{x}^{det}$ corresponds to a score vector of an object proposal (e.g., $R_r$). Then the two matrices are normalized by the softmax layers $\sigma(\cdot)$ along the category direction (column-wise) and proposal direction (row-wise) respectively, leading to $\sigma(\mathbf{x}^{cls})$ and $\sigma(\mathbf{x}^{det})$. Finally, the instance-level classification score for object proposals are computed by the element-wise product $\mathbf{x} = \sigma(\mathbf{x}^{cls}) \odot \sigma(\mathbf{x}^{det})$. The image-level classification score for the $c_{th}$ class is computed as $p_c = \sum_{i=1}^{N} x_{i,c}$. The MIL multiple-label classification loss is used to supervise MIL classification: $\mathcal{L}_{mlc} = -\sum_{c=1}^{C} \{y_c \log p_c + (1 - y_c) \log(1 - p_c)\}$. The proposal scores $\mathbf{x}$ can be used to select proposals as detected objects. However, this step is prone to selecting proposals corresponding to the most discriminative object instances and object parts.

To address this general WSOD issue, [12] introduces multi-stage OICR branches to refine the MIL head. The proposal feature vectors from Proposal Feature Extractor are fed to $K$ refinement stages of OICR instance classifiers. Each $k_{th}$ stage is supervised by the instance-level pseudo-labels selected from the $N_k$ top-scoring proposals in previous stage. Each instance classifier consists of an FC layer and a softmax function, and outputs a proposal score matrix $\mathbf{x}^k \in \mathbb{R}^{(C+1) \times N}$. (The background class is the $(C + 1)_{th}$ category in $\mathbf{x}^k$.). The loss for the $k_{th}$ classifier is defined as $\mathcal{L}_{ref}^k = -\frac{1}{N_k} \sum_{r=1}^{N_k} \sum_{c=1}^{C+1} \hat{y}_{r,c}^k \log x_{r,c}^k$. ($r$ corresponds to the region $R_r$.) During the inference, the proposal score matrices of all $K$ classifiers are summed to predict bounding boxes with Non-Maximum Suppression (NMS). Besides the refinement loss, [40, 8] also incorporate a regression loss to further improve the bounding box localization capability. Following Fast-RCNN, we define $L_{reg}^k = \frac{1}{G^k} \sum_{r=1}^{G^k} smooth_{L1}(t_r, \hat{t}_r)$ where $G^k$ is the total number of positive proposals in $k$-th branch. $t_r$ and $\hat{t}_r$ are tuples including location offsets and sizes of $r$-th predicted and ground truth bounding-box.

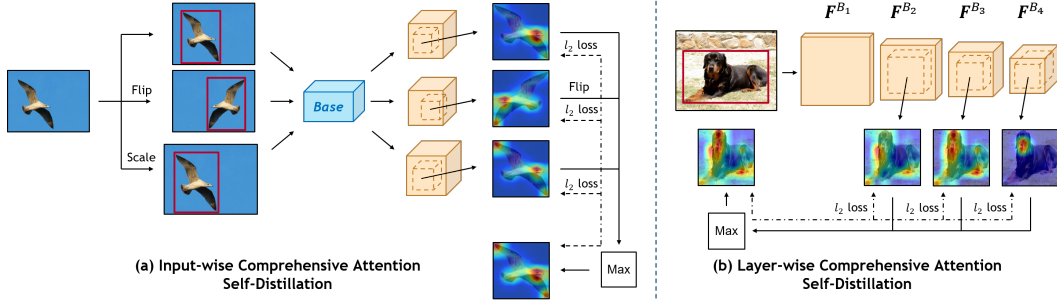

Figure 3: Comprehensive Attention Self-Distillation (CASD). **Input-wise (IW) CASD** in (a) distills the comprehensive attention maps that are aggregated from the proposal feature maps of the same image under different transformations (horizontal flipping and scaling). **Layer-wise (LW) CASD** in (b) computes the comprehensive attention maps over different network layers. Both IW-CASD and LW-CASD encourage the WSOD network to learn balanced and consistent representation.

## 3.2 Comprehensive Attention Distillation

Building upon [12], Comprehensive Attention Self-Distillation (CASD) encourages consistent and balanced representation learning in two folds: Input-wise CASD and Layer-wise CASD.

**Input-wise (IW) CASD.** IW-CASD conducts CASD over input images under multiple transformations (Fig. 3 (a)). Consider a set of inputs including the original image $\mathbf{I}$, its horizontally flipped image $\mathbf{I}^{flip} = T_{flip}(\mathbf{I})$ and its scaled image $\mathbf{I}^{scale} = T_{scale}(I)$. These input images are fed into the same WSOD feature extractor to get the image feature maps outputted by the last convolution layer: $\mathbf{F}$, $\mathbf{F}^{flip}$ and $\mathbf{F}^{scale}$, respectively. At each refinement branch, $N_K$ positive and negative object proposals are selected by the same way as [12]. For the $r_{th}$ object proposal $R_r$ ($r = 1, \cdots, N_K$), the proposal feature $\mathbf{P}_r \in \mathbb{R}^{H \times W \times L}$ is cropped from $\mathbf{F}$, and is used to compute the proposal attention map $\mathbf{A}_r \in \mathbb{R}^{H \times W}$ by channel-wise average pooling and element-wise Sigmoid activation.

$$\mathbf{A}_r(m, n) = S(\frac{1}{L} \sum_{i=1}^{L} \mathbf{P}_r^i(m, n)),  \tag{1}$$

where $\mathbf{A}_r(m, n)$ denotes the proposal attention magnitude at spatial location $(m, n)$ of proposal $R_r$. $S(\cdot)$ is the element-wise Sigmoid operator. The proposal attention maps $\mathbf{A}_r^{flip}$ and $\mathbf{A}_r^{scale}$ from $\mathbf{F}^{flip}$ and $\mathbf{F}^{scale}$ can be computed in the same way as Eq. 1.

The example in Fig. 3 (a) shows that attention maps $\mathbf{A}_r$, $\mathbf{A}_r^{flip}$ and $\mathbf{A}_r^{scale}$ focus on different parts of proposal $R_r$. This is the common case in WSOD: for input image under different transformations, the feature distribution within class does not correlate with the feature distribution between classes. Unless regularized, the image-level classification of WSOD always relies on the most discriminative features, which only correspond to the proposals on salient objects or object parts.

The union of these attention maps (denoted by $\mathbf{A}_r^{IW}$) always covers more comprehensive parts of the object than individual attention maps,

$$\mathbf{A}_r^{IW} = max(\mathbf{A}_r, T_{flip}(\mathbf{A}_r^{flip}), \mathbf{A}_r^{scale}),  \tag{2}$$

where $max(\cdot)$ is the element-wise max operator. We define the input-wise CASD based on $\mathbf{A}_r^{IW}$ as a refinement problem on the WSOD network: updating the WSOD feature extractor such that any transformed variants of the same image should generate comprehensive attention close to $\mathbf{A}_r^{IW}$. To this end, $\mathbf{A}_r^{IW}$ are simultaneously approximated from individual attention maps. For each $k_{th}$ refinement branch , the IW-CASD loss function is defined as

$$\mathcal{L}_{IW}^k = \frac{1}{N_K} \sum_{r=1}^{N_K} \left( \left\| \mathbf{A}_r^{IW} - \mathbf{A}_r \right\|_2 + \left\| \mathbf{A}_r^{IW} - T_{flip}(\mathbf{A}_r^{flip}) \right\|_2 + \left\| \mathbf{A}_r^{IW} - \mathbf{A}_r^{scale}) \right\|_2 \right).  \tag{3}$$

$N_K$ is the total number of selected proposals in the $k_{th}$ branch. The computation of $A_r^{IW}$ and updating of the WSOD feature extractor are alternative steps in training. We consider this is a knowledge distillation process from $A_r^{IW}$ to the WSOD network itself, and hereby name it as self-distillation.

However, the alternating process above may also lead to local optimum. In particular, the individual attention map containing more high-intensity elements may dominate attention maps associated with other transformations. We balance the distillation among different transformations by independently applying Inverted Attention (IA) [26] on each individual attention. For each attention, IA randomly masks out top features highlighted by attention maps and force more features to be activated. This training technique regularizes CASD and produces better results (see ablation study in Section 4).

Additionally, the aggregated proposal score matrices are used in the WSOD network training for each transformed variant of the same image. Specifically, in the MIL head, IW-CASD takes several transformed variants of the same image to generate groups of proposal instance score matrices for each branch ($K + 1$ in total). Within each $k_{th}$ group, each score matrix provides a transformed view of the original image for detection. Thus it is natural for WSOD to aggregate the score matrices within each $k_{th}$ group to form a robust proposal score matrix as $\bar{\mathbf{x}}^k = \frac{1}{3}(\mathbf{x}^k + \mathbf{x}^k_{flip} + \mathbf{x}^k_{scale})$.

**Layer-wise(LW) CASD** (LW-CASD) operates on proposal feature maps of image $\mathbf{I}$ produced at multiple layers of WSOD feature extractor (see Fig. 3 (b)). The feature extractor network consists of $Q$ convolutional blocks $B_1, ..., B_Q$, each of which outputs a feature map $\mathbf{F}^{B_1}, ..., \mathbf{F}^{B_Q}$, respectively. $\mathbf{F}^{B_Q}$ is used as the feature map for generating proposal feature vectors in the MIL head. As illustrated by the example in Fig. 3 (b), at the early layers of a CNN, the network focuses on local low-level features, while at deeper layers, the network tends to focus on global semantic features. Conducting CASD over layers enriches the proposal features with more granularity.

In LW-CASD, we use RoI pooling at different feature blocks to generate proposal feature maps, such that they share the same spatial size. From these proposal attention maps, the layer-wise attention maps $\mathbf{A}^{B_q}$s are generated in a similar way as Eq. 1, and aggregated to obtain the comprehensive attention maps,

$$\mathbf{A}_r^{LW} = max(\mathbf{A}_r^{B_1}, \cdots, \mathbf{A}^{B_Q}), \tag{4}$$

where $max(\cdot)$ is the element-wise max operator. For the $k_{th}$ refinement branch, the IW-CASD loss is

$$\mathcal{L}_{LW}^k = \frac{1}{N_K} \sum_{q=1}^{Q} \sum_{r=1}^{N_K} \left\| \mathbf{A}_r^{LW} - \mathbf{A}_r^{B_q} \right\|_2. \tag{5}$$

The choice of layers in IW-CASD are studied in Section 4.

**Remarks.** 1) Channel-wise average pooling+sigmoid in Eq. 1 is not the only choice for attention estimation. We also explored other mechanisms such as Grad-CAM [19], but there is only a negligible performance difference. Eq. 1 is used due to its simplicity and computation efficiency. 2) Besides flipping and scaling, any other image transformations could also be used in IW-CASD. 3) In Eq. 3 and Eq. 5, gradients are not back-propagated to the comprehensive attention maps $A_r^{IW}$ or $A_r^{LW}$.

**Overall Loss Function.** The overall loss for training a CASD-based WSOD network is composed of the losses from Sec. 3.1 and Sec. 3.2.

$$\mathcal{L}_{CASD-WSOD} = \mathcal{L}_{mlc} + \sum_{k=1}^{K} (\alpha \mathcal{L}_{ref}^k + \beta \mathcal{L}_{reg}^k + \gamma \mathcal{L}_{IW}^k + \gamma \mathcal{L}_{LW}^k), \tag{6}$$

where $\alpha$, $\beta$ and $\gamma$ balance the weights of different losses. $K$ is the number of refinement branches.

## 4   Experimental Results

**Datasets and Metrics.** Three standard WSOD benchmarks, PASCAL VOC 2007, VOC 2012 [41] and MS-COCO [42], are used in our experiments. Both VOC 2007 and VOC 2012 contain 20 object classes with an additional background class. In VOC 2007, the total of $9,962$ images are split into three subsets: $2,501$ for training, $2,510$ for validation and $4,951$ testing. In VOC 2012, all the $22,531$ images are split into the $5,717$ training images, $5,823$ validation images, the rest $10,991$ test images. For both datasets, we followed the standard routine in WSOD [12, 8, 5, 7] to train on the train+val set and evaluate on the test set. For MS-COCO trainval set, the train set ($82,783$ images) is used for training and the val set (40K images) is used for testing. Only image-level labels are utilized in training. For evaluation, a predicted bounding box is considered to be positive if it has an IoU$> 0.5$ with the ground-truth. For VOC, mean Average Precision mAP$_{0.5}$ (IoU threshold at $0.5$) is reported. For MS-COCO, we report mAP$_{0.5}$ and mAP (averaged over IoU thresholds in $[.5 : 0.05 : .95]$).

**Implementation details.** All experiments were implemented in PyTorch. The VGG16 and ResNet50 pre-trained on ImageNet [43] are used as WSOD backbones. Batch size is set to be $T$ that is the number of input transformations. The maximum iteration numbers are set to be $80K$, $160K$ and $200K$ for VOC 2007, VOC 2012, and MS-COCO respectively. The whole WSOD network is optimized in an end-to-end way by stochastic gradient descent (SGD) with a momentum of 0.9, an initial learning rate of 0.001 and a weight decay of 0.0005. The learning rate will decay with a factor of 10 at the $40\text{k}^{th}$, $80\text{k}^{th}$, and $120\text{k}^{th}$ iteration for VOC 2007, VOC 2012 and MS-COCO, respectively. The total number of refinement branches $K$ is set to be 2. The confidence threshold for Non-Maximum Suppression (NMS) is 0.3. For all experiments, we set $\alpha = 0.1$, $\beta = 0.05$ and $\gamma = 0.1$ in the total loss. Note that we reimplemented the OICR loss in Pytorch and found that using a fixed $\alpha = 0.1$ is slightlty better than the adaptive weighting policy in vanilla OICR ($48.9\%$ mAP$_{0.5}$ vs. $48.3\%$ mAP$_{0.5}$ on VOC 2007). Thus we keep the fixed weight $\alpha$ for the OICR loss. Selective Search [11] is used to generate about $2,000$ object proposals for each image.

To have a fair comparison with other methods, following [12, 7], multi-level scaling and horizontal flipping data augmentation are conducted in training. Multi-level scaling is also used in testing. Specifically, in the data augmentation, the short edges of input images are randomly re-scaled to a scale in $\{480, 576, 688, 864, 1200\}$, and the longest image edges are capped to $2,000$, then a random horizontal flipping is randomly conducted on the scaled images. In evaluation, input images are augmented with all five scales.

## 4.1 Ablation Studies

We conducted three sets of ablation studies on the VOC 2007 with metric mAP$_{0.5}$ (%) in Table 1- 3. All results are based on the VGG16 backbone.

**CASD main configurations.** We conducted ablation studies on the main components of CASD in Table 1 under the mAP$_{0.5}$ metric. Our baseline achieves $48.9\%$ mAP$_{0.5}$. With the proposed Input-wise CASD ("+IW"), the baseline is boosted to $54.1\%$ mAP$_{0.5}$ while the proposed Layer-wise CASD (+LW) improves the baseline to $52.3\%$ mAP$_{0.5}$. Both IW-CASD and LW-CASD can consistently improve the baseline with a clear $5.2\%$ and $3.4\%$ gain respectively. Combining IW with LW ("+IW+LW") can further boost the performance to $55.3\%$ mAP$_{0.5}$ that is $6.4\%$ better than the baseline. In addition, IW-CASD without Inverted Attention (IA) and proposal score aggregation (PSA), "+IW w/o IA+PSA", could achieve $51.4\%$ mAP$_{0.5}$ which is $2.5\%$ superior to the baseline. Proposal score aggregation brings a $1.2\%$ mAP$_{0.5}$ performance gain, leading to $52.6\%$ mAP$_{0.5}$ in IW-CASD without Inverted Attention ("+IW w/o IA"). And Inverted Attention (+IW) has a $1.5\%$ mAP$_{0.5}$ performance boost, justifying the effectiveness of IA.

Then we further demonstrate the validity of regression branch and stronger augmentation. With the regression branch, "+IW+LW+Reg" achieves $56.1\%$ mAP$_{0.5}$ that is $0.8\%$ better than "+IW+LW". Moreover, with stronger data augmentation of "flip+scale+color", we can achieve the best CASD performance ("+IW+LW+Reg*") $56.8\%$ mAP$_{0.5}$. Here the "Color" augmentation applies some photo-metric distortions to the images which is the same as those described in [44, 45].

**CASD layer configurations.** Built upon the baseline, additional ablation study on layer configurations of LW-CASD are shown in Table 2. The $B_i$ is the $i_{th}$ convolutional block of VGG16 and $B_4$ denotes the last block before the FC layer for image classification. As shown in the table, the best result $52.3\%$ mAP$_{0.5}$ is obtained with LW-CASD using $B_4 + B_3 + B_2$ blocks. As demonstrated in Fig. 3 (b), these results indicate that middle-level feature attention maps (e.g. $B_2$ and $B_3$) encode balanced discriminative clues (more spatially distributed than $B_1$) for WSOD, while the low-level feature attention maps (e.g. $B_1$) contain noisy spatial information which may deteriorate the performance. Besides Table 2, all other experiments in this work use the $B_4 + B_3 + B_2$ configuration in CASD.

**Attention regularization strategies.** Besides our attention self-distillation, there are other regularization strategies emerged for semi-supervised object detection and multi-label classification such as Prediction Consistency [46] and Attention Consistency [29]. For a thorough demonstration of the advantage of our attention distillation strategy, we implement these strategies under the WSOD setting and compare them with the CASD. Similar to [46], prediction consistency in WSOD is implemented by minimizing the JS-Divergence between predictions of differently transformed data. Attention consistency [29] in WSOD force the attention of the last proposal feature maps to be consistent under different input transformations based on the MSE loss.

| Method | Transformations | VOC 2007 |
|---|---|---|
| Baseline | flip+scale | 48.9 |
| +IW w/o IA+PSA | flip+scale | 51.4 |
| +IW w/o IA | flip+scale | 52.6 |
| +IW | flip+scale | 54.1 |
| +LW | flip+scale | 52.3 |
| +IW+LW | flip+scale | 55.3 |
| +IW+LW+Reg | flip+scale | 56.1 |
| +IW+LW+Reg* | flip+scale+color | **56.8** |

Table 1: Ablation study of CASD main configurations. "IW" as Input-Wise CASD, "LW" as Layer-Wise CASD, "IA" as Inverted Attention, "PSA": proposal score aggregation, "Reg" as regression branch.

| Method | VOC 2007 VOC 2012 |
|---|---|
| WSDDN [5] | 34.8 | - |
| OICR [12] | 41.2 | 37.9 |
| PCL [7] | 43.5 | 40.6 |
| C-MIL [13] | 50.5 | 46.7 |
| WSOD2(+Reg.) [40] | 53.6 | 47.2 |
| Pred Net [47] | 52.9 | 48.4 |
| C-MIDN [15] | 52.6 | 50.2 |
| MIST(+Reg.) [8] | 54.9 | 52.1 |
| CASD(ours) | **56.8** | **53.6** |

Table 4: Comparison with SOTA WSOD results (VGG16 backbone, mAP$_{0.5}$) on the PASCAL VOC 2007 and VOC 2012.

| Method | VOC 2007 |
|---|---|
| $B_4 + B_3$ | 51.8 |
| $B_4 + B_3 + B_2$ | **52.3** |
| $B_4 + B_3 + B_2 + B_1$ | 51.4 |

Table 2: Ablation study of CASD layer configurations based on the Baseline+LW.

| Method | VOC 2007 |
|---|---|
| Prediction Consistency [46] | 50.2 |
| Attention Consistency [29] | 51.0 |
| Attention Distillation (Ours) | **52.6** |

Table 3: Ablation study of strategies for attention regularization based on Baseline+IW w/o IA.

| Method | Backbone | mAP | mAP$_{0.5}$ |
|---|---|---|---|
| PCL [7] | VGG16 | 8.5 | 19.4 |
| C-MIDN [13] | VGG16 | 9.6 | 21.4 |
| WSOD2 [40] | VGG16 | 10.8 | 22.7 |
| MIST(+Reg.) [8] | VGG16 | 11.4 | 24.3 |
| MIST(+Reg.) [8] | ResNet50 | 12.6 | 26.1 |
| CASD(ours) | VGG16 | 12.8 | 26.4 |
| CASD(ours) | ResNet50 | **13.9** | **27.8** |

Table 5: Comparison with SOTA WSOD results on the MS-COCO dataset.

We compared IW-CASD ("+IW w/o IA") with the other two methods in Table 3. IW-CASD is superior to both prediction and attention consistency strategies by a 2.4% mAP$_{0.5}$ and 1.6% mAP$_{0.5}$ performance gain respectively. This validates the superiority of CASD over other attention consistency methods.

**Sensitivity analysis on loss weight $\gamma$:** For the loss weight $\gamma$ in Eq. 6, we evaluated CASD on VOC 2007 with $\gamma = 0.05, 0.075, 0.1, 0.15, 0.2$, and got mAP $54.1\%, 54.7\%, 55.3\%, 55.0\%, 55.0\%$ respectively, which demonstrates that CASD is robust to $\gamma$.

## 4.2 Comparison with the State-of-the-arts

Here we experimentally compare the full version of CASD with other state-of-the-art methods. In Table 4, with the same VGG16 backbone, CASD reaches the new state-of-the-art mAP$_{0.5}$ of 56.8% and 53.6% on VOC 2007 and VOC 2012, which are 1.9% and 1.5% higher than the latest state-of-the-art (MIST(+Reg.) [8]). In the results on MS-COCO (Table 5). With the VGG16 backbone, CASD produces 12.8% mAP and 26.4% mAP$_{0.5}$, outperforming the VGG16 version of MIST(+Reg.) by clear margins of 1.4% and 2.1%. With the ResNet50 backbone, CASD achieves the state-of-the-art 13.9% mAP and 27.8% mAP$_{0.5}$ outperforming MIST(+Reg.) by clear margins of 1.3% mAP and 1.7% mAP$_{0.5}$. Qualitative visualization of detection results can be found in the supplementary file.

## 5 Conclusion

In this paper, we proposed the Comprehensive Attention Self-Distillation (CASD) algorithm to regularize the WSOD training. CASD aggregates "free" resources of spatial supervision within the WSOD network, such as the different attentions produced under multiple image transformations and low-to-high feature layers. Through self-distillation on the WSOD network, CASD enforces instance-balanced and spatially-consistent supervision over objects and achieves the new state-of-the-art WSOD results. As a training module, we believe CASD can be generalized and benefit other weakly-supervised and semi-supervised tasks, such as instance segmentation, pose estimation.

## Broader Impact

This paper pushes the frontier of the weakly supervised object detection and reduce its performance gap with the supervised detection. This work is also a general regularization approach that may benefit semi-supervised learning, weakly-supervised learning, and self-supervised representation learning. The ultimate research vision is to potentially relieve the burden of human annotations on training data. This effort may reduce the cost, balance human bias, accelerate the evolution of machine perception technology, and help us to understand how to enable learning with minimal supervision.

## Acknowledgments and Disclosure of Funding

This work was supported by the Intelligence Advanced Research Projects Activity (IARPA) via Department of Interior/ Interior Business Center (DOI/IBC) contract number D17PC00340.

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
