[Supplementary Material]

# Comprehensive Attention Self-Distillation for Weakly-Supervised Object Detection
## *Supplementary Material*

**Zeyi Huang**[*], **Yang Zou**,[*] **Vijayakumar Bhagavatula, and Dong Huang**
Carnegie Mellon University, Pittsburgh, PA, 15213, USA
{zeyih@andrew, yzou2@andrew, kumar@ece, donghuang@}.cmu.edu

# 1 More Experimental Results

## 1.1 Visualization

Figure 4: Visualization of CASD-WSOD results. **Top**: success cases; **Bottom**: failure cases. Only the attention maps of high confidence proposals by WSOD detection are overlaid on input images. Green bounding boxes denote the ground truth.

---

[*]The authors contributed equally.

Figure 5: Comparison of MIST [1] and CASD. **Top**: MIST detection; **Middle**: CASD detection; **Bottom**: CASD overlaid with attentions.

We first present qualitative results of CASD-WSOD in Fig. 4. Recall that WSOD conducts classification on object proposals (e.g., bounding boxes generated by Selective Search [2]) with image-level class labels. The object proposals receive high classification scores are considered as objects detected by WSOD. In Fig. 4, only the attention maps of high confidence proposals by WSOD detection are overlaid on input images.

Fig. 4 shows both the success and the failure cases of CASD. When the objects are relatively big and are completely visible, CASD tends to succeed. The failure cases mainly occur on objects that are small or under heavy occlusion. We deduce that there are two main factors contributing to the failure: (1) The Selective Search (SS) [2] algorithm may not generate good proposals for heavily occluded objects. This could be improved by using better objectness proposal generators such as the RPN of Faster-RCNN. (2) The incomplete appearance of the occluded objects make CASD difficult to learn the long-range dependency among the object parts. This could be improved by hard-sample mining in CASD training.

Fig. 5 compares results of MIST [1] and CASD. CASD has better success in detecting high-quality bounding boxes than MIST. This localization advantages of CASD benefit from its learning of comprehensive attention (see the bottom row of Fig. 5). We further demonstrate the localization quality of CASD in Table 6.

## 1.2 CorLoc on Trainval Sets

Correct Localization (CorLoc) was used in some previous works to evaluate performance on the VOC 2007 and VOC 2012 trainval sets. CorLoc only evaluates the localization accuracy of detectors. *In all those works, CorLoc was reported when WSOD is trained on both the training and validation sets, and tested on both sets.* This is probably why CorLoc was mostly used for ablation, not for the main comparison between algorithms.

For completeness, we provide this additional metric in Table 6. CASD has the best overall localization accuracy among all compared methods.

| Method | VOC 2007 | VOC 2012 |
|---|---|---|
| ***WSOD State-of-the-Art*** | | |
| WSDDN [3] | 53.5 | - |
| OICR [4] | 60.6 | 62.1 |
| PCL [5] | 62.7 | 63.2 |
| C-MIL [6] | 65.0 | 67.4 |
| WSOD2(+Reg.) [7] | 69.5 | 71.9 |
| Pred Net [8] | **70.9** | 69.5 |
| C-MIDN [9] | 68.7 | 71.2 |
| MIST(+Reg.) [1] | 68.8 | 70.9 |
| ***Ours*** | | |
| CASD | 70.4 | **72.3** |

Table 6: Comparison of the localization performance (CorLoc) on the VOC 2007 and VOC 2012 trainval sets.

## 1.3 More Ablation Studies and Clarification

**Different $\gamma$s for $L_{IW}$ and $L_{LW}$:** In Eq. 6, we have a single loss weight $\gamma$ for both the IW-CASD loss $\mathcal{L}_{IW}^k$ and LW-CASD loss $\mathcal{L}_{LW}^k$. Another policy is to set different loss weights for the two losses respectively. Which way is better? So we conduct the following ablation study on VOC 2007. Fixing $\gamma_{IW} = 0.1$, CASD gets $55.0\%, 55.5\%, 55.3\%, 54.8\%, 54.8\%$ mAP$_{0.5}$ when $\gamma_{LW} = 0.05, 0.075, 0.1, 0.15, 0.2$ respectively. Fixing $\gamma_{LW} = 0.1$, CASD gets $54.3\%, 54.6\%, 55.3\%, 55.1\%, 54.9\%$ mAP$_{0.5}$ when $\gamma_{IW} = 0.05, 0.075, 0.1, 0.15, 0.2$ respectively. At $L_{IW} = 0.1$ and $L_{LW} = 0.075$, CASD achieves $55.5\%$ mAP$_{0.5}$ which is only $0.2\%$ better than $55.3\%$ mAP$_{0.5}$ (the best performance of single $\gamma$). Thus we conclude that single $\gamma$ is a good trade-off between performance and hyper-parameter tuning.

**Evidence for consistency and completeness in CASD:** First, the attention maps and predicted bounding boxes in Fig. 1 (b) of the main paper and Fig. 5 compare the results of OICR/MIST and CASD, qualitatively demonstrating CASD gets more consistent and complete object features. Second, on the horizontal flipped VOC 2007 test set, CASD achieves $56.5\%$ mAP$_{0.5}$ which is similar to $56.8\%$ of the unflipped test set. This indicates that CASD is consistent w.r.t flipping.

**CASD with Grad-CAM:** In CASD, we utilize channel-wise average pooling+sigmoid to get the attention map. We conduct an ablation study on layer-wise CASD with Grad-CAM that gets $52.0\%$ mAP$_{0.5}$ on VOC 2007. This is slightly worse than $52.6\%$ mAP$_{0.5}$ of layer-wise CASD with channel-wise average pooling+sigmoid. Thus the latter is computationally more efficient and adopted in our paper.

**Clarification on the branch number of $k$ in OICR:** The vanilla OICR [4] has 3 OICR branches ($k = 3$), and suggests the larger $k$ the better results. We only use $k = 2$ in all our experiments due to our GPU limitations. Thus CASD may achieve better results by setting $k = 3$ on GPU with sufficient memory.