[Reviews · NeurIPS 2020]

Review 1

Summary and Contributions: This paper solves the problem of weakly-supervised object detection. It presents a Comprehensive Attention Self-Distillation (CASD) mechanism, which is developed from a previous work OICR. The main contribution is the introduction of the input-wise CASD and the layer-wise CASD, both of which aim to capture integral object attentions given the proposals.

Strengths: 1) The motivation of this paper is well demonstrated and the claims sound. It points out the key issues, which have not been solved by previous work. 2) The novelty of this paper is fine. Based on the MIL framework, this paper proposes to introduce both input-wise CASD and layer-wise CASD to encourage the neural network to generate integral object attentions. 3) The experimental results are good. Compared to most recently published work (e.g., in CVPR'19, ICCV'19, and CVPR'20), this work performs much better on the PASCAL VOC and COCO benchmarks. 4) Relevance to the NeurIPS community. The attention mechanism proposed in this paper, I think, is useful for the NeurIPS community. It provides a promising way to generate integral object attentions. 5) The literature review is also good. I know more fully-supervised object detection methods than the weakly-supervised ones. However, after reading the paper, I do understand the working mechanisms in most weakly-supervised appraoches.

Weaknesses: 1) The motivation part in the introduction section should be rewritten. The authors point out the shortcomings of most previous work and then say that attention-based learning method is a good candidate for solving these issues. So, why does attention-based method work in these situations? The authors should give some brief descriptions of the advantages of attention-based methods. 2) The motivation of this paper is strong but the novelty of the contribution is a bit weak. However, based on the good results, I think it is fine. 3) In Table 2, it is good to see which layers contributes more to the final performance. The authors in the corresponding paragraph (starting from L255) explains that attentions maps from low layers have noisy spatial information. According to my knowledge, this may be due to the fact lower layers in CNNs encodes more detail information and are short of capturing highly semantic information to distinguish the semantic objects. It would be better if the authors could visualize the resultant attention maps to see whether this is the case. 4) The English of this paper needs to be improved. For instance, in L136, a 'to' should be inserted before the word 'select' and in L228, what is MS-CsOCO? The authors should carefully revise the manuscript. Overall speaking, this paper is well written except for some typos. After going through the rebuttal, I found that most of my concerns have been solved. I also read the reviews from other reviewers and found that some concerns do exist. However, I do not think these concerns affect the quality of this paper much. The novelty of this paper is significant, so I think it deserves a good submission.

Correctness: Yes

Clarity: Yes

Relation to Prior Work: Yes

Reproducibility: Yes

Additional Feedback: See the weaknesses.


Review 2

Summary and Contributions: This WSOD (weakly-supervised object detection) paper builds upon the prior works, WSDDN and OICR, by introducing the additional indirect supervision that requires the model to produce consistent feature maps with respect to (1) input transformations (flip, scale) and (2) different layers. The results are reported on Pascal and COCO detection.

Strengths: 1. Novelty and significance Introduction of consistency requirements for weak supervision looks like a promising research direction. The consistency criterion has not been introduced in any prior weakly-supervised computer vision tasks, as far as I can tell. It would be nice to see more work in weakly-supervised localization methods utilizing the consistency loss which can be built for free. 2. Interesting categorization/taxonomy of issues in WSOD. The illustration and categorization of WSOD failure cases are interesting (missing instance, clustered instances, and part domination). Part domination has been widely known in the weakly-supervised learning domain, while the other two are not (and thus interesting). They seem fairly plausible and the above transformation-consistency criterion is likely to alleviate the issues. One catch is that the observation is not experimentally verified over a dataset. In the next version of paper, please include a more in-depth analysis of the issues.

Weaknesses: 1. Many, many hyperparameters. Separation of validation and test sets? Many weakly-supervised X methods fail to work in real world, despite the promises in the research papers. Why is that? (1) Typically they have overfit to the test set. (2) Introduction of many, many hyperparameters in the method. (3) They have implicitly introduced full supervision through an extensive hyperparameter search over a massive amount of fully supervised samples. [A] provides a good diagnosis of the above issues in the context of WSOL (localization instead of detection). [A] further argues that when the methods proposed in the time frame 2016-2019 are re-evaluated under the same amount of implicit full supervision (hyperparameter search with the same architecture, same data, and the same amount of computational resources), no WSOL method since the vanilla CAM (CVPR 2016) has truly made an advance. I am worried that this may also be the case for WSOD. I observe many bad signals in this paper. For example, there are many, many hyperparameters in this method. - Choice of Selective Search as the proposal method (e.g. why not EdgeBox or MCG?) - Hyperparameters in the proposal method (e.g. hyperparameters controlling the size and frequency of objects). - K in the OICR method. - Method for score aggregation (L145) - NMS hyperparameter (L146, L230) - The choice of scaling and flipping as input transformations (why not others?) - Why max in L169? (why not mean or min?) - Use of Inverted Attention (L182). - Choice of layers (L203). - alpha=0.1, beta=0.05, and gamma=0.1 in Equation 6 & L230. Along with a long list of hyperparameters, the paper does not properly describe how the choices are made. Are they decided through validation? If so, do you separate the validation set (for hyperparameter search) and the test set (for final results & comparison against prior methods) ? For Tables 1-5, please indicate whether the results are based on val or test set. If not decided through validation, do you have enough evidence that the decided set of hyperparameters is expected to transfer well to unseen data and network architectures? Is there any empirical evidence? If so, is there any conceptual argumentation why the hyperparameters transfer? Answering those questions is not only of academic interest but also of practical interest. Only by fully answering to those questions can a WSOD method be verified to be beating the prior arts and become truly useful in real applications. 2. Performance is bounded by the proposal method. Relying on hand-crafted proposals for detection is old-fashioned. For end-to-end trained deep object detectors, no one is really using proposals anymore. The reported performances are thus severely limited by the proposal qualities and are far below the sensible level of detection performance these days. At least to put those numbers in perspective, it would be nice to also report the oracle WSOD performance, given the proposal method. [A] Choe et al. Evaluating Weakly Supervised Object Localization Methods Right. CVPR 2020.

Correctness: It is quite likely that the validation & hyperparameter selection are not correctly done. Please address the weakness #1 above.

Clarity: There are some inclarities in certain parts. For example, OICR is not clearly described at the technical level though it is the central building block for the proposed method. Please add more details for - "generated from he proposal scores" (L141) - "aggregated" (L145) - "incorporate a regression loss" (L147) And add citation to WSDDN in the paragraph L128-137.

Relation to Prior Work: The work is based on WSDDN and OICR. The updates from those methods are described well.

Reproducibility: No

Additional Feedback: prospective -> perspective fine-grain -> fine-grained a FC layer -> an FC layer \Phi -> \mathrm{sigmoid} Describe what L is in L156 CsOCO -> COCO ***POST REBUTTAL*** From the positive point of view, the paper did a good job at incorporating a less-explored cue for weak supervision: equivariance (though with a pinch of salt - equivariance was already explored in "old-school" WSOD - e.g. Bilen et al. Weakly Supervised Object Detection with Posterior Regularization. BMVC'16) > Unlike WSOL, WSOD field has better separated validation set from the evaluation set. (e.g. using VOC07 test for validating design choices and using VOC12 test and COCO val for evaluation). I believe this is a much better practice than what's happening in WSOL. My apologies for overlooking this key difference. On my own defense, however, the authors should have included those key implementation details in the paper. The paper only describes the dataset statistics and does not define the use of splits and where the intermediate and final results are reported. There is one remaining issue. In WSOL, the evaluation problem was twofold. (1) Validating on the test set. (2) Even after introducing the validation set, you still need to make design choices with implicit full supervision (by validating your choices with GT bounding boxes in validation set). The first problem is addressed by the authors' careful evaluation protocol. The second problem is only partially addressed, by saying that the chosen hyperparameters and design choices transfer across (VOC07,VOC12) and (VOC07,COCO), implying that there's no need for the use of bounding-box-annotated validation set for every test set. I'm only lukewarm about this argument. You still need the fully-supervised validation set for VOC07 in this case and it's questionable if you can say you're "only using the image-level category labels" (Line 2) to do object detection. It is also questionable if the found hyperparameters transfer to other datasets with different object categories and statistics. Having said this, I believe it's unfair to accuse this single paper of the dubious evaluation practice that has persisted in the field for years. Given that the authors have at least address the val-test split issue well, I'd like to raise my rating to acceptance and require the authors to include their evaluation protocol in the paper. I think the paper is valuable otherwise - providing a good taxonomy and review of recent WSOD methods and exploring the equivariance for weak supervision. > Regarding large number of hyperparameters, authors claim that most of these hyperparameters are inherited from the baseline and only a small part of the hyperparameters are introduced along with proposed CASD and fairly validated in WSOD benchmarks. Yes, it is unfair to turn a criticism against the whole field into the one against a single paper. For the practical point of view, however, the practice of inheriting HPs from previous papers significantly hinders the performance improvements over years (the result of this is the use of selective search in WSOD in 2020 ;) ), but this lets authors successfully defend their case. All in all, I'm in for accepting this paper, but would like to send clear message that it's time to think about - How to set up the WSOD task: should we acknowledge the use of full supervision? - How to set up evaluations to show that HPs transfer? (e.g. by introducing new benchmarks on OpenImages?) - How to motivate papers to use more advanced components (e.g. replacing selective search with RPN).


Review 3

Summary and Contributions: This paper proposes a comprehensive attention self-distillation method for weakly supervised object detection. The method enforces consistent attention among feature maps from images with different transformations and different convolutional layers, which ensures weakly supervised object detectors generating tight object boxes. Experimental results on PASCAL VOC and COCO show that the proposed method obtains the state-of-the-art weakly supervised object detection results.

Strengths: 1. The proposed method makes sense and has not been studied in previous weakly supervised object detection works. 2. Very promising results are obtained on PASCAL VOC and COCO datasets.

Weaknesses: To me, there are no significant weaknesses related to this work. Here are some minor weaknesses. 1. Experiments. It would be better to conduct more ablation experiments to analyze the proposed method. 1) From Line 189-190, the authors stated that classification scores from images with different transformations are aggregated to generate supervision for OICR [10], which should also benefit to OICR training. The authors should show results of only using aggregated scores (without L_{IW} and L_{LW}) and only using L_{IW} (without aggregated scores) to show the performance gains of each component. 2. There are four hyper-parameters (three loss weights and the number of refinement branches) in Eq. (6). The loss weight alpha and the number of refinement branches K are different from the original setting in OICR [10]. It would be better to show results of different hyper-parameter value choices. 2. Writing. There are some unclear descriptions in this paper. 1) For L_{IW} and L_{LW}, it is unclear which parts will receive gradients during back-propagation. From my understanding, I suppose A_{r}^{IW} and A_{r}^{LW} do not receive gradients. I'm not sure whether I'm right or not. 2) In Eq. (3), A^{IW} should be A_{r}_{IW}. 3) In Eq. (6), L_{reg}^{k} has not been defined and introduced. 3. Why the authors do not use a single A_{r} = max(A_{r}^{IW}, A_{r}^{LW}) for both L_{IW} and L_{LW}?

Correctness: Yes. The claims and the empirical methodology should be correct.

Clarity: The paper written could be improved, see the Weakness part.

Relation to Prior Work: Most of prior works are clearly discussed. There are some prior works also generate attention maps by channel-wise average pooling [a, b, c]. It would be better to refer to these works when introducing how to generate attention maps (line 157). [a] Incorporating network built-in priors in weakly-supervised semantic segmentation. TPAMI, 2018 [b] Weakly Supervised Region Proposal Network and Object Detection. ECCV, 2018 [c] Attention-based Dropout Layer for Weakly Supervised Object Localization. CVPR, 2019

Reproducibility: Yes

Additional Feedback: This paper focuses on the most important issues in current weakly supervised object detection works and proposes a good solution. The authors have addressed most of our concerns in their rebuttal. Therefore, I would keep my original accept rating. The authors should revise their paper in their final version according to reviews.


Review 4

Summary and Contributions: The paper describes an approach to improve the training of Weakly Supervised Object Detection networks (WSOD). It builds off of a previous method called Online Iterative Classification Refinement (OICR) and adds a novel method called Comprehensive/Complementary Attention Self-Distillation (CASD). CASD aims to solve several issues that are common with WSOD detectors such as missing inconspicuous instances, grouping objects in cluttered scenes and focusing on localizing discriminative object parts as opposed to the entire object. CASD achieves this by computing input-wise and layer-wise attention maps, combining them respectively, and performing self-distillation on the attention maps . The input-wise attention map is computed by first transforming the input image (identity, flip, scale, color etc.), passing the transformed images through a feature extractor, computing the attention map (average on tensor depth and elementwise sigmoid on spatial dimensions) and finally combining them by computing the element wise max across these attention maps. Each of the input-wise attention maps is then trained to match the merged attention map via an L2 loss. The layer-wise attention map is computed by extracting feature maps from intermediate layers, performing ROI pooling to ensure consistent spatial dimensions and computing a merged attention map in a similar manner to the input-wise attention map. Each of the layer-wise attention maps is then trained to match the merged attention map via an L2 loss. Finally, the network is trained via a weighted loss consisting of the attention losses, classification losses, regression losses and MIL classification loss used in the refinement branches. The paper's main contribution is an attention distillation approach that claims to address several existing problems with WSOD methods and achieves a new state of the art.

Strengths: The method described in the paper is well motivated and the paper does a good job of explaining how the method attempts to solve existing problems with WSODs. Each design choice is justified and explained well. In addition, the paper includes several details about OICR/MIL that is important for contextualizing the contribution of the method and for reproduction of the implementation. The empirical baselines are comprehensive and fair. The authors present models with two different backbones (VGG16 & ResNet50) that better match the baseline models to measure the effect of the method alone. The paper also includes some useful ablation studies on layer configuration, data augmentation and attention modes as well. The authors also claim that this approach establishes a new state of the art.

Weaknesses: While the application of attention self-distillation to WSOD appears to be novel, there doesn't seem to be a substantially novel contribution in terms of the methodology. Picking the hyperparameters for the attention layer could potentially be quite challenging since the degenerate solution of uniform values minimizes the attention loss component. The authors mention on line 230 that they used values of alpha=0.1, beta=0.05 and gamma=0.1 but did not describe how they picked those values. Furthermore, the choice to set the same gamma value for both attention components is not explained. An ablation study on this would be quite beneficial in determining the effect of varying the values here. Since the authors claim state of the art performance, it would be more convincing if they included their code in the supplementary material so that the reviewers can verify that the results are reproducible.

Correctness: The authors claim that CASD addresses the 3 issues they mention in the introductory paragraph. Furthermore, they state on line 101 that "Moreover, existing methods for other tasks only encourage consistency of features, rather than the completeness of features. Specifically, the features tend to focus on object parts but fail to localize less salient objects in WSOD. CASD encourages both consistent and spatially complete feature learning guided by the comprehensive attention maps, which explicitly addresses WSOD issues above". However there is no qualitative or quantitative evidence provided that convincingly shows that CASD explicitly solves these cases better than existing WSOD. In fact on line 7 of the supplementary material, the authors state: "When the objects are relatively big and are completely visible, CASD tends to succeed. The failure cases mainly occur on objects that are small or under heavy occlusion". This appears to contradict the claim that CASD addresses the case where "the salient object is detected while inconspicuous instances tend to be ignored." Upon taking a closer look at the qualitative comparisons provided in Figure 2 of the appendix, it looks like CASD still suffers from "part domination". The leftmost example detects the person's torso while the middle example detects the horse's head (with high attention activation). Similar comments apply for the inconsistent WSOD argument. It would be more convincing to have some experiments that show the performance of CASD on transformed images.

Clarity: The title and abstract describe the approach as "Comprehensive Attention Self-Distillation", the title of section 3.2 is "Complementary Attention Distillation" and the approach is referred to as "Complementary Attention Self-Distillation (CASD)" at multiple places including figures. The abstract states on line 3 that "WSOD detectors are prone to detect bounding boxes on salient objects, cluttered objects and discriminative object parts". These do not sound more like strengths than weaknesses of WSODs. The paragraph on line 33 describes the actual problems in a clearer manner. The paper states on line 204: "We also explored other class-wise attention mechanisms such as Grad-CAM [17], but there is only a negligible performance difference." It would be good to include details like this in the supplementary material. The paper is missing a few details regarding implementation. First, for VOC, is the model trained only on the training data or the trainval split? Secondly, the numbers of the splits in for VOC 2007 do not add up ( 4092+2501+2510 = 9103 != 9962) Could the authors please clarify if this was a typo or whether some images were omitted? On line 267, Prediction Consistency is cited as [41], and Attention Consistency is cited as [23] but in Table 3, they are flipped. The paper also has grammatical errors at several places and phrasing could be improved. Overall, while the paper contains most of the information about the paper, there are several inconsistencies, missing or incorrect details and the writing could generally be improved.

Relation to Prior Work: The paper does a good job of discussing prior work, contextualizing its contribution and including fair baselines to previous competitive approaches. While the paper omits references to a few related recent works, it is still fairly comprehensive. Omitted recent related works in WSOD: - SLV: Spatial Likelihood Voting for Weakly Supervised Object Detection by Chen et. al (CVPR 2020) - Boosting Weakly Supervised Object Detection with Progressive Knowledge Transfer. Zhong et. al. ECCV 2020

Reproducibility: Yes

Additional Feedback: While the paper describes a promising approach that claims state of the art results, and has a few strengths, there are some issues that should be resolved before being ready for publication. === After rebuttal === The paper is improved with the additional sensitivity analysis on \gamma and ablation study. Contingent upon fixing the inconcistencies, writing issues and including missing details in the paper, I have updated the score to "Marginally above the acceptance threshold".

[Author Response · NeurIPS 2020]

We thank all reviewers and AC for the time and constructive comments. Below we address the main concerns, and will
fix other issues/typos and cite suggested references in the revised paper. Code will be released upon acceptance.

**R2 / R3 / R4: Sensitivity analysis on** $\gamma$**:** Evaluated on VOC07, CASD with $\gamma = 0.05, 0.075, 0.1, 0.15, 0.2$ gets mAP
$54.1\%, 54.7\%, 55.3\%, 55.0\%, 55.0\%$ respectively, demonstrating CASD is robust to $\gamma$.

**R1: Advantages of attention-based method:** In Line 41-55 and Fig 1, we motivate our attention-based method by
visualizing the features activated by WSOD networks. We observe that failures in WSOD are closely associated with the
flawed features. Non-attention-based WSOD methods typically improve pseudo-labels and loss functions, instead of the
features. In contrast, our method explicitly uses the structural info in attention maps, and promotes discriminative and
consistent features on whole objects (Fig 1). Our method is orthogonal and applicable to non-attention-based WSOD.

**Visualization of layer-wise attention maps:** Fig. 3 (b) visualizes the attention maps of 2nd/3rd/4th conv blocks.

**R2: Data splits:** We follow the data splits in [7,8,10,11,13,34,42]: For VOC 2007 and 2012, we train on the train+val
set and evaluate on the test set. For COCO, we train on the train set and evaluate on the val set.

**Clarification on hyper-parameter selection:** We agree that over-tuning hyper-parameters does not improve WSOD
research. However, as long as hyper-parameters are database-independent, certain choices of networks, parameters
and settings represent reasonable assumptions required to solve the ill-posed WSOD problem. Examples of the above
hyper-parameters include those in selective search, object proposal, score aggregation, NMS, scaling and flipping.
CASD inherits above hyper-parameters from PCL and OICR and fixes them on VOC07, VOC12 and COCO.

For the remaining CASD-specific hyper-parameters (Inverted Attention, choice of layers, max operation, loss weight $\gamma$),
we follow the routine of [7,8,10,11,13,34,42], and only tune CASD parameters on VOC07 with the VOC07 test set, and
then keep them fixed for VOC12 and COCO. The state-of-the-art results on VOC12 and COCO demonstrate that the
VOC07 hyper-parameters of CASD transfer well and are practical for new databases.

We share the reviewer's concern that some WSOL methods in [A] lack clear validation. However, our work and
[7,8,10,11,13,34,42] all select hyper-parameters on VOC07 and fix them for VOC12 and COCO. The VOC07 results,
although over-tuned on the test set, are still valuable for comparing performance "upper bounds" among WSOD
methods. The VOC12 and COCO results are NOT tuned on their test sets.

**Performance bounded by the proposal method:** We agree that using RPN in WSOD should beat Selective Search
(SS). However, CASD is a training module independent of proposal methods. We adopt the same SS as most WSOD
literatures only for fair comparisons. We will add both RPN-based CASD and Faster-RCNN results in the revised paper.

**R3: IW-CASD without classification score aggregation:** Without score aggregation and IA, IW-CASD achieves
$51.4\%$ mAP, clearly improving the baseline by $2.5\%$ mAP. Adding score aggregation brings $1.2\%$ mAP improvement.

**Clarification on** $k$ **and** $\alpha$**:** The vanilla OICR [10] has 3 OICR branches ($k = 3$), and suggests the larger $k$ the better
results. We only use $k = 2$ in all our experiments due to our GPU limitations. We reimplemented the OICR loss in
Pytorch by ourselves. On VOC07, the OICR baseline at $\alpha = 0.1$ achieves $48.9\%$ which is slightly superior to $48.3\%$
using the adaptive weighting policy in vanilla OICR. Thus we keep the fixed weight for the OICR loss.

**Clarification on method: (1).** For $L_{IW}$ and $L_{LW}$, the gradients are not back-propagated to the comprehensive
attention maps $A_r^{IW}$ and $A_r^{LW}$. **(2).** In Eq. (4), $A^{IW}$ should be $A_r^{IW}$. **(3).** Following Fast-RCNN, we define
$L_{reg}^k = \sum_{r=1,...,G^k} smooth_{L1}(t_r, \hat{t}_r)/G^k$, $G^k$ is the total number of positive proposals in $k$-th branch. $t_r$ and $\hat{t}_r$ are
tuples including location offsets and sizes of $r$-th predicted and gt bounding-box. **(4).** We will adopt the notation of
$A_r = max(A_r^{IW}, A_r^{LW})$ in revised paper. Thanks for the suggestion.

**R4:** We sincerely thank the detailed feedback. We will fix the typos, inconsistency, add details and improve writing.
**Different** $\gamma$**s for** $L_{IW}$ **and** $L_{LW}$**:** We conduct the following ablation study on VOC07. Fixing $\gamma_{IW} = 0.1$, CASD
gets $55.0\%, 55.5\%, 55.3\%, 54.8\%, 54.8\%$ when $\gamma_{LW} = 0.05, 0.075, 0.1, 0.15, 0.2$ respectively. Fixing $\gamma_{LW} = 0.1$,
CASD gets $54.3\%, 54.6\%, 55.3\%, 55.1\%, 54.9\%$ when $\gamma_{IW} = 0.05, 0.075, 0.1, 0.15, 0.2$ respectively. At $L_{IW} = 0.1$
and $L_{LW} = 0.075$, CASD achieves $55.5\%$ which is only $0.2\%$ better than $55.3\%$ (the best performance of single $\gamma$).
Thus we conclude that single $\gamma$ is a good trade-off between performance and hyper-parameter tuning.

**Evidence for consistency and completeness:** First, the attention maps in Fig. 1 b) compare the results of baseline
OICR and CASD, qualitatively demonstrating CASD gets more consistent and complete object features. More examples
will be added in the revised paper. Second, on the horizontal flipped VOC07 test set, CASD achieves $56.5\%$ mAP
which is similar to $56.8\%$ of the unflipped test set. This indicates that CASD is consistent w.r.t flipping.

**Result with Grad-CAM:** The layer-wise CASD with Grad-CAM gets $52.0\%$ mAP on VOC07. This is slightly worse
than $52.6\%$ mAP of layer-wise CASD with channel-wise average pooling+sigmoid. The latter is computationally more
efficient and adopted in our paper.

[Meta-Review · NeurIPS 2020]

This paper proposes a Comprehensive Attention Self-Distillation (CASD) training method for tackling weakly-supervised object detection. The method is empirically assessed on COCO and PASCAL VOC benchmarks where it shows impressive performance. A common concern across all reviews was the issue with number of hyper-parameters of the method. Reviewers found detailed authors' response very helpful and some increased their initial scores. Another common concern, but less critical, was about writing, which can be improved for the final version. Overall, the paper is seen as an important contribution and thus I recommend accept.